# Multicolour synthesis in lanthanide-doped nanocrystals through cation exchange in water

Sanyang Han[1], Xian Qin[2], Zhongfu An[3], Yihan Zhu[4], Liangliang Liang[1], Yu Han[4], Wei Huang[3,5] & Xiaogang Liu[1,2,6]

Meeting the high demand for lanthanide-doped luminescent nanocrystals across a broad range of fields hinges upon the development of a robust synthetic protocol that provides rapid, just-in-time nanocrystal preparation. However, to date, almost all lanthanide-doped luminescent nanomaterials have relied on direct synthesis requiring stringent controls over crystal nucleation and growth at elevated temperatures. Here we demonstrate the use of a cation exchange strategy for expeditiously accessing large classes of such nanocrystals. By combining the process of cation exchange with energy migration, the luminescence properties of the nanocrystals can be easily tuned while preserving the size, morphology and crystal phase of the initial nanocrystal template. This post-synthesis strategy enables us to achieve upconversion luminescence in $Ce^{3+}$ and $Mn^{2+}$-activated hexagonal-phased nanocrystals, opening a gateway towards applications ranging from chemical sensing to anti-counterfeiting.

[1] Department of Chemistry, National University of Singapore, Singapore 117543, Singapore. [2] Institute of Materials Research and Engineering, Agency for Science, Technology and Research, Singapore 117602, Singapore. [3] Key Laboratory of Flexible Electronics & Institute of Advanced Materials, Jiangsu National Synergetic Innovation Center for Advanced Materials, Nanjing Tech University, Nanjing 211816, China. [4] Advanced Membrane and Porous Materials Center, Physical Science and Engineering Division, King Abdullah University of Science and Technology, Thuwal 23955-6900, Saudi Arabia. [5] Key Laboratory for Organic Electronics and Information Displays & Institute of Advanced Materials, Jiangsu National Synergetic Innovation Center for Advanced Materials, Nanjing University of Posts and Telecommunications, Nanjing 210023, China. [6] SZU-NUS Collaborative Innovation Center for Optoelectronic Science & Technology, Key Laboratory of Optoelectronic Devices and Systems of Ministry of Education and Guangdong Province, College of Optoelectronic Engineering, Shenzhen University, Shenzhen 518060, China. Correspondence and requests for materials should be addressed to Y.H. (email: yu.han@kaust.edu.sa) or to W.H. (email: iamwhuang@njtech.edu.cn) or to X.L. (email: chmlx@nus.edu.sg).

With the rapid development of nanoscience and nanotechnology, lanthanide-doped upconversion nano-crystals[1–5] have recently emerged as an important class of luminescent materials, owing to their potential applications ranging from biological imaging[6–8] and multiplexing sensing[9–11] to security encoding[12–14] and volumetric display[15]. Despite significant progress made, the vast majority of approaches for making upconversion nanocrystals have involved *de novo* synthetic techniques such as hydrothermal reaction[16,17], co-precipitation[18–20] and thermal decomposition[21–23]. To access different colour emissions[24–26], one has to perform a new set of reactions and require stringent control over a variety of experimental conditions, including the amount of dopant precursors and surfactants, solvent type, reaction time and temperature. This practice is clearly time-consuming and resource-intensive, and often leads to variation in particle size, phase and morphology[16,20].

Cation exchange reactions at the nanoscale have recently emerged as a powerful tool for controlling composition and phase in colloidal semiconductor nanocrystals[27–31]. These reactions present an alternative solution for modulating emission colours in upconversion nanocrystals. However, different from the band-gap luminescence nature of quantum dots[32–36], the emission from the upconversion nanocrystals stems directly from the lanthanides infused in the host lattice[37–41]. It is important to note that realizing efficient upconversion luminescence typically requires the homogeneous placement of sensitizer and activator ions in rather close proximity, as is the case for $NaYF_4$ nanoparticles co-doped with $Yb^{3+}$ and $Er^{3+}$ (ref. 4). Although a high doping concentration of $Yb^{3+}$ theoretically favours luminescence enhancement[42–44], upconversion nanocrystals with a large $Yb^{3+}$ content (for example, $NaYbF_4$) are highly sensitive to the concentration quenching effect that depletes excitation energy and thus suppresses luminescence. This dilemma makes the cation exchange strategy practically unsuitable for emission colour modulation using conventional host materials (for example, $NaYF_4$, $NaLuF_4$ and $NaYbF_4$; refs 45–48).

It has been well-established that $Gd^{3+}$-based host materials could effectively bridge the gap of energy transfer from sensitizers to activators through long-range energy migration in the sub-lattice[24,41]. Because of its large energy gap ($\sim 4.0$ eV) between the ground state ($^8S_{7/2}$) and the lowest excited state ($^6P_{7/2}$), the $Gd^{3+}$ ion also serves as an ideal energy reservoir to suppress the concentration quenching of sensitized luminescence in crystalline nanophosphors.

Here we reason that utilization of a $Gd^{3+}$-based host lattice may leverage multicolour synthesis in upconversion nanocrystals through cation exchange under mild conditions. By making use of myriad energy transfer pathways between dopant ions, our approach proves useful for accessing a plethora of optical nanomaterials of uniform size, shape and phase (Fig. 1). In particular, we achieve upconversion emission from $Ce^{3+}$ or $Mn^{2+}$ doped in hexagonal-phased nanocrystals. This allows us to generate a record long-lived luminescence of $\sim 600$ ms for $Mn^{2+}$-activated nanocrystals.

## Results

**Synthesis and characterization.** In a typical procedure, hexagonal phase $NaGdF_4$:Yb/Tm@$NaGdF_4$ core-shell nanocrystals were firstly synthesized as a model system by a co-precipitation procedure (Supplementary Fig. 1; ref. 24). Subsequently, surface-bound oleic acid molecules were removed by the treatment of HCl to generate ligand-free nanocrystals (Supplementary Figs 2 and 3). Cation exchange was then induced by mixing an aqueous solution containing a $TbCl_3$ precursor with the as-prepared colloidal sample under ambient conditions for 1 h. High-resolution transmission electron microscopic (TEM) imaging reveals the single-crystalline hexagonal structure of the resulting nanocrystals after cation exchange (Fig. 2a and Supplementary Fig. 4). Low-resolution TEM imaging and the size distribution analysis of the nanocrystals before and after cation exchange show no obvious changes in the particle size and morphology (Fig. 2b and Supplementary Figs 5–7). In addition, X-ray diffraction of the samples confirms

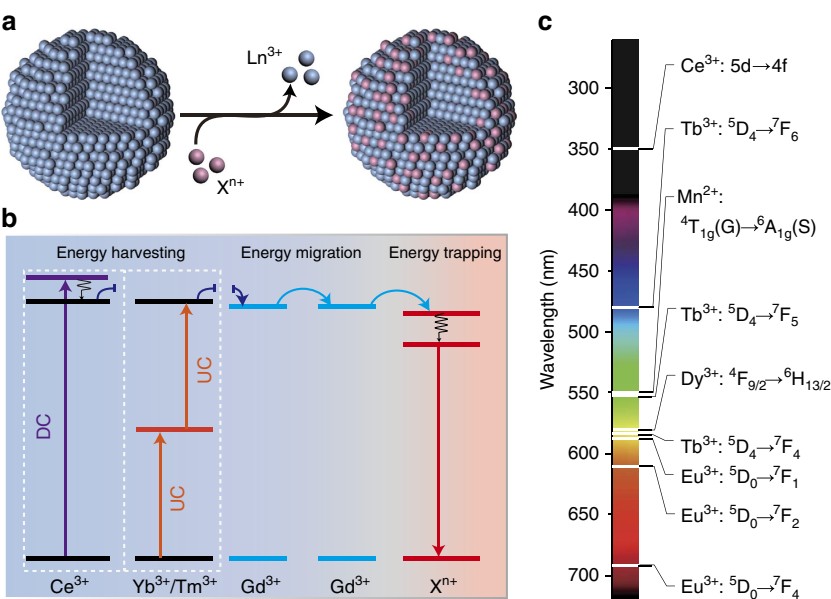

**Figure 1 | Rational design for emission tuning in lanthanide-doped nanocrystals through cation exchange.** (**a**) Schematic representation of a typical cation exchange process, occurring at the particle surface, between lanthanide ($Ln^{3+}$) and exchange ($X^{n+}$) ions. (**b**) Proposed energy management process for cation exchange-mediated luminescence tuning in the nanocrystals. The energy transfer process mainly comprises energy harvesting, energy migration and energy trapping through different types of lanthanides. DC and UC represent downconversion and upconversion processes, respectively. (**c**) Typical luminescent ions used for cation exchange-mediated luminescence tuning and their main emitting transitions in the ultraviolet and visible part of the electromagnetic spectrum.

that the hexagonal phase is completely preserved after the post-synthetic treatment (Fig. 2c, Supplementary Figs 8 and 9 and Supplementary Note 1).

Electron energy loss spectroscopy analysis on a single nanoparticle reveals that the $Tb^{3+}$ ions are mainly located within the outmost layers of the core-shell nanocrystals (Fig. 2d–f and Supplementary Fig. 10). To further substantiate the occurrence of cation exchange, we carried out inductively coupled plasma mass spectroscopy analysis of the colloidal sample after cation exchange. For a series of experiments, we found that the amount of $Gd^{3+}$ ions discharged from the core-shell nanocrystals increases with increasing $Tb^{3+}$ concentration (Supplementary Figs 11–14 and Supplementary Note 2).

**Spectroscopic study of cation-exchanged nanocrystals.** Through $Gd^{3+}$-mediated energy migration in the core-shell structure, the excitation energy could be efficiently transferred from the Yb/Tm pair in the core layer to the activator ions in the shell layers upon cation exchange (Fig. 3a and Supplementary Figs 15 and 16).

The cation exchange process can be visualized by monitoring the emission colour change of the colloidal samples before and after addition of the $Tb^{3+}$ or $Eu^{3+}$ precursor (Fig. 3b and Supplementary Movies 1 and 2). The luminescence spectra of the samples measured at room temperature show an increase in $Tb^{3+}$ or $Eu^{3+}$ emission intensity but a decrease in $Gd^{3+}$ intensity with increasing concentrations of the exchange ions (Supplementary Figs 17 and 18).

The cation exchange process is generally controlled by three parameters: reaction time, temperature, and the ion concentration used for exchange in the solution (Supplementary Note 3). In our study, $Tb^{3+}$-exchanged nanocrystals were taken as an example to study the influence of these three parameters on optical properties. We first investigated the time-dependent cation exchange at ambient conditions by monitoring the emission of a colloidal sample ($NaGdF_4$:Yb/Tm@$NaGdF_4$; 26.2 mg) after addition of $TbCl_3$ (20 μmol). As shown in Fig. 3c, the intensity of $Tb^{3+}$ emission gradually increased over time and reached a plateau after 8 min, at which time the cation exchange reached a dynamic equilibrium (Supplementary

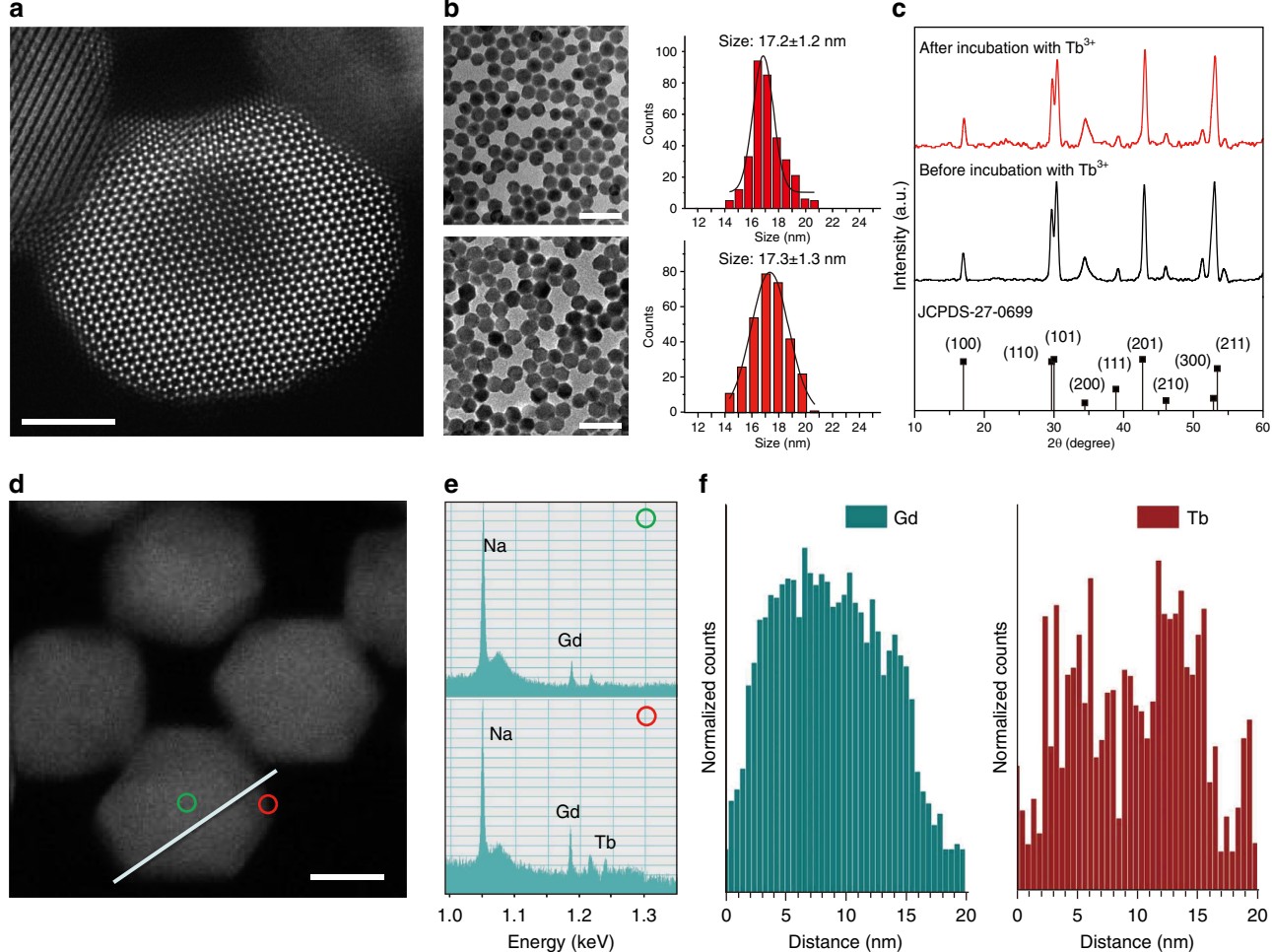

**Figure 2 | Structural characterization of NaGdF₄:Yb/Tm@NaGdF₄ nanoparticles before and after cation exchange.** (**a**) Typical high-resolution transmission electron microscopic (TEM) image of a nanoparticle treated with $TbCl_3$ at room temperature. (**b**) Low-resolution TEM images and corresponding size distributions of the as-prepared NaGdF₄:Yb/Tm@NaGdF₄ nanoparticles, obtained before (top panel) and after (bottom panel) incubation with $TbCl_3$. (**c**) Corresponding powder X-ray diffraction patterns of the core-shell nanocrystals. Note that all peaks can be well indexed in accordance with hexagonal-phase NaGdF₄ structure (Joint Committee on Powder Diffraction Standards file number 27-0699). (**d**) Typical scanning transmission electron microscopic (STEM) image of the nanoparticles treated with $TbCl_3$. (**e**) Electron energy loss spectroscopy (EELS) point analysis collected, respectively, from green and red circle marked in **d**. (**f**) EELS line profile recorded by scanning along the white line shown in **d**. Note that both elemental analyses in **e**,**f** reveal that more Tb content is present at particle outer layer, while more Gd content exists at particle inner layer. Scale bar, 5 nm for panel **a**, scale bar, 50 nm for panel **b**, scale bar, 10 nm for panel **d**.

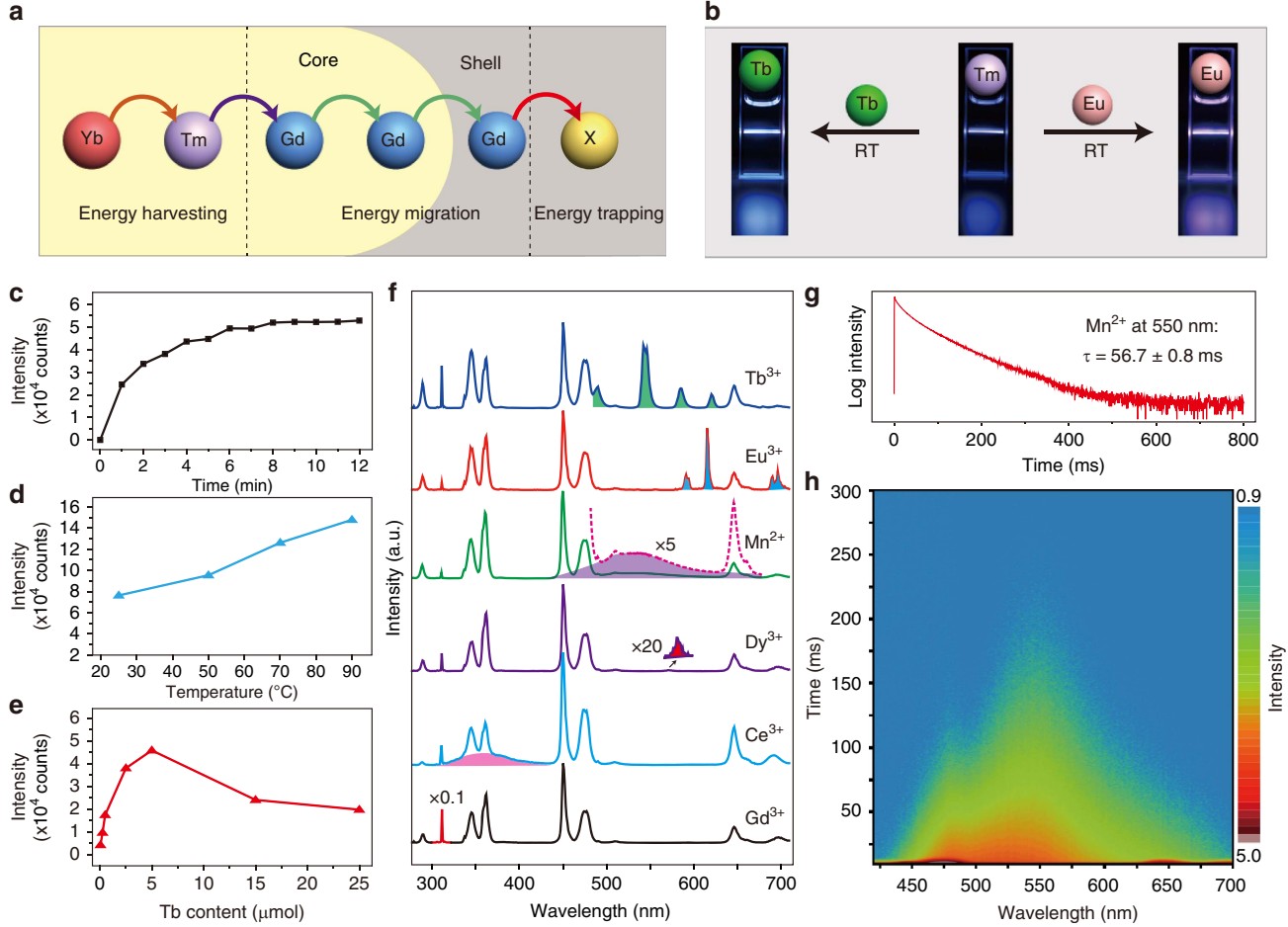

**Figure 3 | Optical investigation of Gd-based upconversion nanocrystals through cation exchange. (a)** Schematic representation of energy transfer process in $NaGdF_4:Yb/Tm@NaGdF_4$ nanocrystals after cation exchange with an activator ion (denoted as X). **(b)** Photoluminescence images showing the change in the emission colour of $NaGdF_4:Yb/Tm@NaGdF_4$ colloidal solutions upon addition of $Tb^{3+}$ or $Eu^{3+}$ ions at room temperature. **(c–e)** Emission intensity profiles of the nanocrystal solution measured as a function of cation exchange time, reaction temperature and $Tb^{3+}$ concentration, respectively. Note that the emission of $Tb^{3+}$ at 546 nm is used for intensity measurement. **(f)** Typical photoluminescence spectra of the nanocrystals treated with $TbCl_3$, $EuCl_3$, $MnCl_2$, $DyCl_3$ and $CeCl_3$, respectively. Note that the activator emissions are highlighted with colour. All spectra were recorded under the irradiation of a 980 nm laser with a pump power of 1 W. **(g)** Lifetime decay curve of $Mn^{2+}$ emission at 550 nm from the $MnCl_2$-treated nanocrystals. **(h)** A transient photoluminescence decay image of the $MnCl_2$-treated nanocrystals. The colour change from red to blue indicates the decrease in emission intensity.

Fig. 19). We further performed the controls at different temperatures, and found that the emission can be greatly enhanced by increasing reaction temperature (Fig. 3d and Supplementary Fig. 20), indicating that high temperatures favour the cation exchange process. The emission intensity can also be boosted by slightly enriching the concentration of the exchange ions (Fig. 3e). The optimal concentration of $Tb^{3+}$ for maximum particle emission was estimated to 5 mM (Supplementary Fig. 21).

Based on the above-mentioned optimization of reaction conditions, we successfully prepared $Gd^{3+}$-based upconversion nanocrystals containing various types of activators ($Eu^{3+}$, $Dy^{3+}$, $Ce^{3+}$, $Mn^{2+}$) by the cation exchange approach (Fig. 3f, Supplementary Fig. 22, Supplementary Table 1 and Supplementary Note 4). It is worth noting that the upconversion emission of $Ce^{3+}$ and $Mn^{2+}$ is observed for the first time in hexagonal-phased $NaGdF_4$ host materials (Supplementary Figs 23–27). Significantly, we obtained a record long-lived $Mn^{2+}$ luminescence of $\sim 600$ ms (lifetime: $\sim 56.7$ ms) (Fig. 3g). The long decay time of $Mn^{2+}$ emission can be used to resolve the spectral overlapping issue between $Mn^{2+}$ and $Tm^{3+}$ emissions by the time-gated spectroscopy (Fig. 3h). Interestingly, an emission

colour change from cyan to green could be discerned by the naked eye on switching off of the 980-nm excitation source (Supplementary Movie 3).

**Mechanistic investigation.** The observation of optical emissions in cation-exchanged nanocrystals and inductively coupled plasma mass spectroscopy analysis reveal that the cation exchange process is dependent critically upon the nature of the exchanged ion; for example, ionic radius and valence charge (Supplementary Fig. 28). To facilitate the exchange of the $Gd^{3+}$ host lattice with $Mn^{2+}$, an elevated temperature of 90 °C is needed to overcome the charge imbalance and lattice strain due to the size mismatch. To understand the cation exchange process, we carried out first-principles calculations by estimating the formation energies of hexagonal-phased $NaGdF_4$ nanocrystals doped with various ions (Supplementary Fig. 29 and Supplementary Table 2; refs 49,50). Our calculations indicate that lanthanides can readily replace Gd atoms at ambient conditions. By comparison, doping of Mn atoms into the $NaGdF_4$ lattice at room temperature requires a large excess of energies ($\sim 1.82$ eV). We further

investigated the charge transfer within the doped nanocrystals by employing charge distribution analysis. It was found that the amount of charge transfer from the doped lanthanides (Eu, Dy, Ce, Tb) to the surrounding F atoms is comparable to that between Gd and F atoms. However, the calculated amount of charge transfer from Mn to F decreased by about $0.5448e$ relative to that between Gd and F, giving rise to reduced dipole polarizability and increased formation energy in Mn-doped nanocrystals. In addition, the significant difference in ionic size between $Mn^{2+}$ and $Gd^{3+}$ ($\sim 13.8$ pm) is another major factor that hinders the cation-exchange process.

Apart from the enhancement of cation exchange at the particle surface, the thermal fluctuation at a high temperature is likely to accelerate the diffusion of the exchanged ions from a particle's surface to its inner region, so that an improved luminescence from the exchanged ions can be observed[51,52]. To confirm our hypothesis, we first mixed aqueous solutions of $TbCl_3$ and $NaGdF_4:Yb/Tm@NaGdF_4$ at room temperature and then isolated the $Tb^{3+}$-modified nanocrystals. Subsequently, we heated the

nanocrystals at 90 °C for 1 h. As anticipated, we observed an increase in $Tb^{3+}$ emission after heat treatment (Supplementary Fig. 30). Our calculations also suggest that the diffusion of doped ions ($Tb^{3+}$, $Eu^{3+}$, $Dy^{3+}$, $Ce^{3+}$ and $Mn^{2+}$) in the $NaGdF_4$ lattice can occur with ease (Supplementary Fig. 31). Taken together, these data indicate that the cation exchange and diffusion at elevated temperatures synergistically contribute to the increased emission of the exchanged cations (Supplementary Fig. 32 and Supplementary Note 5).

We next explored the applicability of this cation exchange approach to colour-tuning in other upconversion and downconversion systems. For instance, we observed an emission colour change for $NaYbF_4:Tb@NaTbF_4$ upconversion nanocrystals from green to yellow upon cation exchange of $Tb^{3+}$ with $Eu^{3+}$ at the particle's surface (Fig. 4a, Supplementary Figs 33–35 and Supplementary Note 6). In the cases where a downconversion process was demonstrated, the substitution of $Gd^{3+}$ in $NaGdF_4:Ce@NaGdF_4$ nanocrystals by a series of activators ($Tb^{3+}$, $Eu^{3+}$, $Dy^{3+}$, $Mn^{2+}$) resulted in

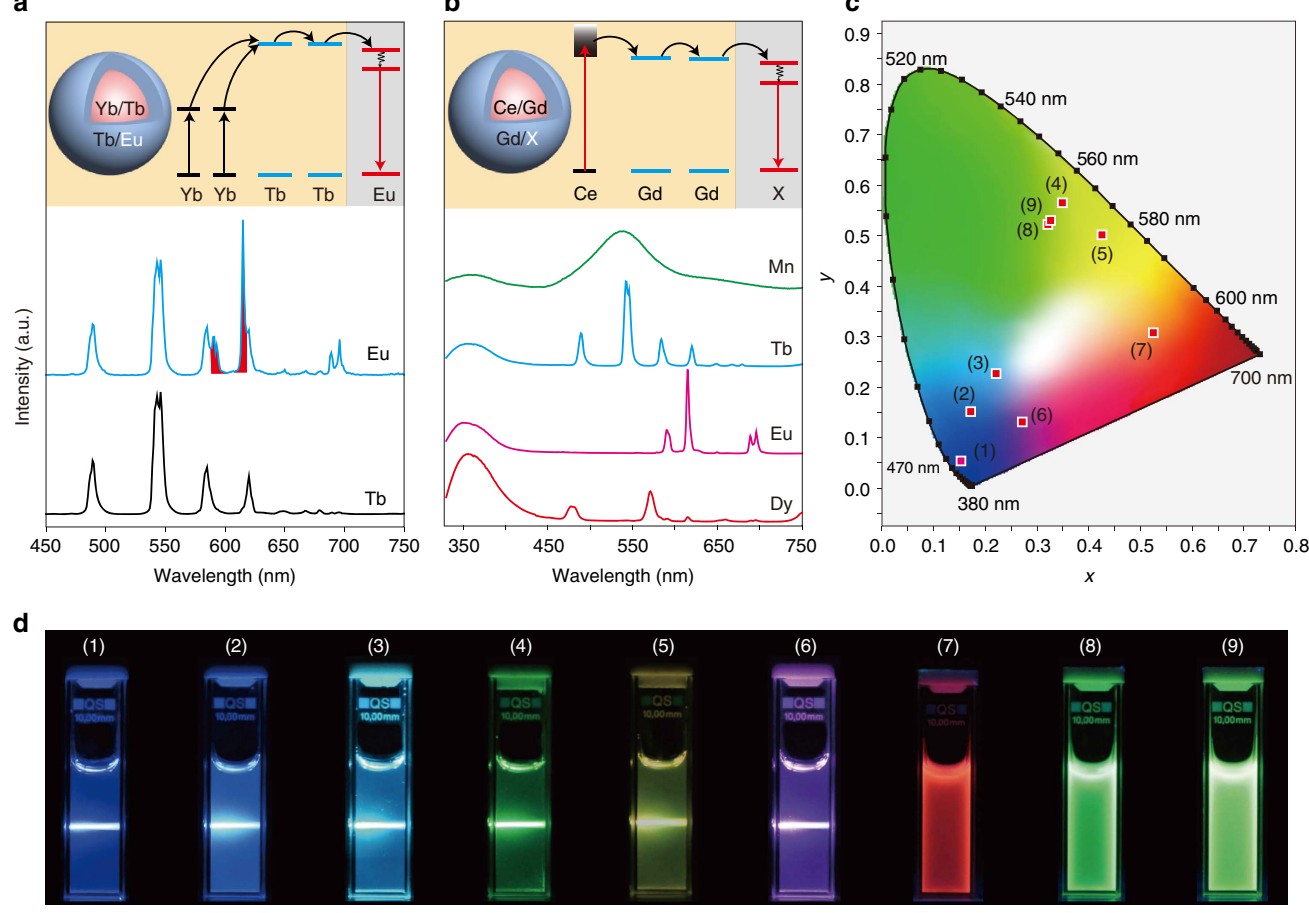

**Figure 4 | Multicolour synthesis in lanthanide-doped nanocrystals through cation exchange.** (**a**) Upconversion luminescence spectra of $NaYbF_4:Tb@NaTbF_4$ nanocrystals before and after treatment with $EuCl_3$. Note that the $Eu^{3+}$ emissions are highlighted in red. Inset shows the proposed energy transfer process in $Eu^{3+}$-treated $NaYbF_4:Tb@NaTbF_4$ nanocrystals. Under the 980 nm excitation, a cooperative upconversion process firstly takes place in the $Yb^{3+}/Tb^{3+}$ pair, followed by an energy migration process in the Tb sub-lattice. The excited energy is subsequently trapped by $Eu^{3+}$ ions. (**b**) Downconversion luminescence spectra of $NaGdF_4:Ce@NaGdF_4$ nanocrystals treated with $TbCl_3$, $EuCl_3$, $MnCl_2$ and $DyCl_3$. Inset shows the corresponding energy transfer mechanism in the nanocrystals. Under the ultraviolet excitation, the excited state of $Ce^{3+}$ is firstly populated. Subsequently, an energy migration process in the Gd sub-lattice bridges energy transfer from $Ce^{3+}$ to exchange ions ($X = Tb^{3+}$, $Eu^{3+}$, $Mn^{2+}$ or $Dy^{3+}$). (**c**) Commission Internationale de l'Eclairage (CIE) chromaticity coordinates of the emissions measured for the as-synthesized colloidal solutions. (**d**) Corresponding luminescence photographs of the colloidal solutions. Samples 1–9 are $NaGdF_4:Yb/Tm@NaGdF_4$, $Mn^{2+}$-exchanged $NaGdF_4:Yb/Tm@NaGdF_4$, $Tb^{3+}$-exchanged $NaGdF_4:Yb/Tm@NaGdF_4$, $NaYbF_4:Tb@NaTbF_4$, $Eu^{3+}$-exchanged $NaYbF_4:Tb@NaTbF_4$, $Eu^{3+}$-exchanged $NaGdF_4:Yb/Tm@NaGdF_4$, $Eu^{3+}$-exchanged $NaGdF_4:Ce@NaGdF_4$, $Mn^{2+}$-exchanged $NaGdF_4:Ce@NaGdF_4$, and $Tb^{3+}$-exchanged $NaGdF_4:Ce@NaGdF_4$, respectively. Note that the luminescence photographs were taken under laser excitation at 980 nm for samples 1–6 and under ultraviolet lamp irradiation at 254 nm for samples 7–9.

$Ce^{3+}$-sensitized multicolour emissions (Fig. 4b, Supplementary Fig. 36 and Supplementary Note 7). Importantly, the capability of our approach to modulating emission colours in lanthanide-doped nanocrystals allowed rapid access to a myriad of different colour spaces with minimum sample processing time (Fig. 4c,d).

## Discussion

The combination of cation exchange and energy migration in lanthanide-doped nanocrystals enables us to precisely tailor the luminescence to the colours of interest, providing the possibility to achieve long-lived luminescence in unexplored regimes of lifetime and without concerning variation in particle size, phase and morphology. As with any methodology, there are seemingly obvious drawbacks. For example, it requires a subset of lanthanide ions capable of participating in the energy migration. However, our data suggest that the advantages of the presented approach far outweigh its limitations. In particular, our approach allows the development of a general, green protocol for preparing multicolour nanoprobes, combining efficient and rapid sample synthesis with significantly reduced solvent and reagent consumption. As such it is anticipated that this technique will greatly expand the repertoire of possible upconversion nanomaterials, with relevant applications for fields as diverse as chemical sensing, biological imaging, photodynamic therapy and anti-counterfeiting.

## Methods

**Nanocrystal synthesis.** Cation-exchanged nanocrystals were prepared by incubating aqueous solution containing ligand-free template nanocrystals and activator ions at ambient conditions for 1 h. Additional experimental details are provided in the Supplementary Methods.

**Characterization.** Ultraviolet–visible spectra were measured on a SHIMADZU ultraviolet-3600 spectrophotometer. Fourier transform infrared spectroscopy spectra were recorded on a Varian 3100 fourier transform infrared spectrometer. Low-resolution TEM images were taken on a JEOL-1400 transmission electron microscope operating at an acceleration voltage of 100 kV. Scanning electron microscopy was carried out on a FEI NOVA NanoSEM 230 scanning electron microscope operated at 5 kV. Powder X-ray diffraction data was obtained on a Siemens D5005 X-ray diffractometer with Cu K$\alpha$ radiation ($\lambda = 1.5406$ Å). The upconversion luminescence spectra were recorded in an Edinburgh FSP920 equipped with a photomultiplier, in conjunction with 980 nm diode laser and a xenon arc lamp (Xe900). The measurement of luminescence lifetime was conducted using a lifetime spectrometer (FSP920, Edinburgh) equipped with a microsecond flash lamp as the excitation source. Upconversion luminescence microscopic images were obtained on an Olympus BX51 microscope with a Nikon DS-Ri1 imaging system adapted to a 980 nm diode laser. Digital photographs were taken with a Nikon D700 camera.

**Data availability.** The authors declare that the data that support the findings of this study are available within the article and its Supplementary Information files. All other relevant data are available from the corresponding author upon request.

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

## Acknowledgements

This work is supported by the Singapore Ministry of Education (Grant R143000627112, R143000642112), Agency for Science, Technology and Research (A*STAR) under the contracts of 122-PSE-0014 and 1231AFG028 (Singapore), National Research Foundation, Prime Minister's Office, Singapore under its Competitive Research Program (CRP Award No. NRF-CRP15-2015-03), National Basic Research Program of China (973 Program, Grant 2015CB932200), National Natural Science Foundation of China (61136003), and the CAS/SAFEA International Partnership Program for Creative Research Teams. Y.H. is grateful to KAUST Global Collaborative Research for the Academic Excellence Alliance (AEA) fund. We thank B. Zhou, Q. Sun, R. Deng, X. Xie, H. Xu and Q. Su for technical assistance and helpful discussion.

## Author contributions

S.H., W.H. and X.L. conceived the project and wrote the paper. S.H. performed the nanocrystal synthesis and luminescence measurements. X.Q. performed the DFT calculation. Y.Z. and Y.H. contributed to the high-resolution TEM imaging and analysis. S.H., Z.A., L.L., Y.H., W.H. and X.L. provided input into the design of the experiments and the preparation of the manuscript.

## Additional information

**Competing financial interests:** The authors declare no competing financial interests.

