## [Peer Review File · Nature Communications]

Reviewers' comments:

Reviewer #1 (Remarks to the Author):

Han et al reports here a facile fabrication of multicolor upconversion nanocrystals through the combination approach of cation exchange in materials chemistry and energy migration in photonics physics. This work is highly innovative and practical for precisely tailoring the luminescence colors of interest, as well as providing a much larger dynamic range for lifetime based multiplexing through the emission of Mn²⁺ ions. The post-synthetic treatment has been approved by the authors to be highly versatile for both photon upconversion and down-conversion applications. More importantly, they show the Mn²⁺-activated hexagonal-phase nanocrystals for the first time. I support its publication after some minor corrections.

1. Some relevant literature on the cation exchange methods for the fabrication of the core-shell upconversion nanomaterials (Dong et al. Chem. Mater., 2012, 24 (7), 1297-1305; Deng et al. Nano Research, 2014, 7, 782-793) should be discussed in the introduction section.
2. On page 1, the second paragraph needs some further explanations: why the surface quenching effect of high Yb³⁺-doped nanomaterials make the cation exchange strategy unsuitable for these host materials including NaYF₄, NaLuF₄, and NaYbF₄? Also, the authors reported that "Because of its large energy gap (~4.0 eV) between the ground state 8S_{7/2}) and the lowest excited state (6P_{7/2}), Gd³⁺ ion also serves as an ideal energy reservoir to suppress the surface quenching of sensitized luminescence in crystalline nanophosphors." Why the NaGdF₄ as the host materials could suppress the surface quenching of the sensitized luminescence?
3. If the optimization of the reaction time is around 8 min according to Figure 3c, why the reaction time of 1 hour was used in the supplementary section? The luminescence intensity is above 5*10⁴ after 8 min incubation (Fig 3c), why the same data in figure S14 is below 4.5*10⁴?
4. One of the emission bands of Eu doped materials was shown as 690nm in Fig S19, but 689 nm in Fig S17 and S32. Also, For Tb, the peak was shown at 489 and 541 nm in Fig S17, at 490 and 543 nm in figure S18b, and at 489 and 542 nm in figure S14. It is better to keep consistent to avoid the confusedness.

Reviewer #2 (Remarks to the Author):

Han et al. reports on the synthesis and investigation of NaGdF₄:Yb/Tm@NaGdF₄ core shell NP's doped with additional Tb ions through (most likely) cation exchange. This is very professional designed, brilliant written report. However from the scientific point of view , to my understanding, there is not much new. Nanoparticles showing anti-stokes emission upon upconversion are not new. As the authors themselves stated the core-shell strategy with or without energy migration has been exploited for different purposes and even tuning of the composition upon cation exchange has been shown at the nanoscale more than ten years ago (ref. 27-31). Thus the referee is wondering where are the new physical insights? That the upconversion emission between Ce³⁺ and Mn²⁺ in hexagonal-phased host material was obtained for the first time is perhaps true. To the referees opinion this does not justify publication in Nat. Commun. The obtainments made by the authors could simply be based on dissolution of the as-prepared NPs in (the still HCl containing solution) followed by growth of a Tb containing shell. Can the authors exclude this? Did they try to find Cl in the NPs besides Tb? To the referees opinion this work cannot be accepted at this stage for publication in a high ranked journal.

On page two the authors said that "To further substantiate the occurrence of cation exchange, we carried out inductively coupled plasma mass spectroscopy (ICP-MS) analysis of the colloidal sample" Supplementary Fig.

11). So my question is. Did they investigated the solution or the "solid" sample? After ion exchange Gd ions should be dissolved in the solution. rejection is recommended

Reviewer #3 (Remarks to the Author):

A. With the burgeoning of actual and potential applications, upconversion nanoparticles (UCNPs) are among the most studied nano-objects presently. Several improvements have been made to their design in order to boost their quantum efficiency and their versatility. Yet, the synthesis of these nanoparticles remains involved and necessitate very strict control of experimental conditions rendering it time-consuming and, also, not easily reproducible. The merit of this work is to propose an elegant and simple way of designing series of UCNPs from an initial batch of gadolinium-based nanoparticles into which different lanthanide ions may be incorporated by cation-exchange under mild conditions. A proof of concept is given for Tb(III) for which reaction time, temperature and ion concentration in the exchange solutions have been optimized. Next, the procedure is applied to three other trivalent lanthanide ions and to Mn(II), allowing the authors to demonstrate UC for the latter and Ce(III) in hexagonal NaGdF₄ nanocrystals for the first time.

B. Novelty. What is novel in the work is well described in the ms. The authors make use of their previous know-how on NaGdF₄ nanocrystals, in particular the fact that Gd(III) can act as energy reservoir and promotes energy migration, as well as described ion-exchange techniques in the synthesis of colloidal semiconductor nanocrystals. The merit of the authors is to have combined two concepts from two different fields to come up with the described results.

C. Altogether, the work has been conducted with great care. All necessary control experiments are presented, and experimental uncertainties are given. This is another high-quality work from this laboratory. I also note that the paper is not restricted to experimental results. First-principle calculations have been carried out to decipher the cation exchange mechanism, its energetics, and charge-transfer within the nanocrystals.

D. If the size, morphology and composition of the synthesized UCNPs are described adequately, nothing is indicated in the text or Supp. Mat. About the reproducibility of these parameters between different batches.

E. The conclusions are adequately supported by the data provided and should be very useful to scientists in the field.

F. As suggested improvements, I would like to see reproducibility data (see D above), as well as some comments/preliminary results on the applicability of the method to other types of UCNPs. Note: Figure S7: average sizes should not be given with so many significant digits, e.g. 21.34+-1.71 nm should read 21.3+-1.7 nm

G. References are appropriate

H. The ms is well organized and well written.

Point-by-Point Response to Reviewers (Manuscript # NCOMMS-16-13467-T)

Reviewer #1:

Han et al reports here a facile fabrication of multicolor upconversion nanocrystals through the combination approach of cation exchange in materials chemistry and energy migration in photonics physics. This work is highly innovative and practical for precisely tailoring the luminescence colors of interest, as well as providing a much larger dynamic range for lifetime based multiplexing through the emission of Mn^{2+} ions. The post-synthetic treatment has been approved by the authors to be highly versatile for both photon upconversion and down-conversion applications. More importantly, they show the Mn^{2+} -activated hexagonal-phase nanocrystals for the first time. I support its publication after some minor corrections.

Question 1: Some relevant literature on the cation exchange methods for the fabrication of the core-shell upconversion nanomaterials (Dong et al. Chem. Mater., 2012, 24 (7), 1297-1305; Deng et al. Nano Research, 2014, 7, 782-793) should be discussed in the introduction section.

Response: It should be noted that the previous works mentioned by this reviewer only focus on the composition tuning by core-shell engineering. In our work, the focus is on the systematic investigation of emission color modulation. Nonetheless, we have added the relevant literatures in the revised manuscript (see ref. 48 and 49).

Question 2: On page 1, the second paragraph needs some further explanations: why the surface quenching effect of high Yb^{3+} -doped nanomaterials make the cation exchange strategy unsuitable for these host materials including NaYF_4 , NaLuF_4 , and NaYbF_4 ?

Response: We thank the reviewer for pointing out this issue, largely due to our unclear writing. It should be the concentration quenching effect rather than the surface quenching effect.

For efficient upconversion to proceed, the sensitizers and activators should be placed in a close proximity in the nanoparticles. Upon a cation exchange reaction with a precursor solution containing activators, the activators are mainly located at the particle's surface. That makes sensitizers (Yb^{3+}) far away from the surface-exchanged activators, thus resulting in inefficient energy transfer. To overcome this problem, one can increase the doping concentration of Yb^{3+} ions in the nanoparticles. Unfortunately, at a high doping concentration of Yb^{3+} in the nanoparticles, the concentration quenching effect among Yb^{3+} ion severely induces a significant dissipation of the excitation energy. Therefore, these issues make the cation exchange strategy unsuitable for conventional NaYF_4 , NaLuF_4 , and NaYbF_4 host materials.

As suggested, we have revised the second paragraph to "upconversion nanocrystals with a large Yb^{3+} content (e.g., NaYbF_4) are highly sensitive to the concentration quenching effect that..." in the revised manuscript for clarity.

Question 3: Also, the authors reported that "Because of its large energy gap (~ 4.0 eV) between the ground state ($^8\text{S}_{7/2}$) and the lowest excited state ($^6\text{P}_{7/2}$), Gd^{3+} ion also serves as an ideal energy reservoir to suppress the surface quenching of sensitized luminescence in crystalline nanophosphors." Why the NaGdF_4 as the host materials could suppress the surface quenching of the sensitized luminescence?

Response: Once again, it should be the concentration quenching rather than the surface quenching. Based on the concentration quenching effect mentioned in Question 2, Gd^{3+} can effectively bridge the energy transfer from Yb/Tm pairs to surface-exchanged activators. This allows us to decrease the Yb^{3+} doping concentration to suppress the concentration quenching effect. The large energy gap of the Gd^{3+} ions can minimize the non-radiative decay and store the excitation energy in its excited states.

Thus, we revised the sentence of " *Gd^{3+} ion also serves as an ideal energy reservoir to suppress the surface quenching of sensitized luminescence in crystalline nanophosphors.*" in last version to be " *Gd^{3+} ion also serves*

as an ideal energy reservoir to suppress the concentration quenching of sensitized luminescence in crystalline nanophosphors.” in the revised manuscript.

Question 4: If the optimization of the reaction time is around 8 min according to Figure 3c, why the reaction time of 1 hour was used in the supplementary section? The luminescence intensity is above 5×10^4 after 8 min incubation (Fig 3c), why the same data in figure S14 is below 4.5×10^4 ?

Response: We thank this reviewer for careful reading of our manuscript. In Figure 3c, we used Tb^{3+} ions as an example to demonstrate that the cation exchange between $NaGdF_4:Yb/Tm@NaGdF_4$ nanoparticles with Tb^{3+} is strongly dependent on the reaction time. Indeed, from this figure, it can be found that the optimal reaction time is around 8 min. But as stated in the manuscript, the cation exchange reaction also depends on the reaction temperature and ion concentration. In addition, the cation exchange process is critically dependent on the nature of the exchange ion (such as ionic radius and valence charge). For other ions (such as Mn^{2+}) that are used for cation exchange with Gd-based nanoparticles, it requires much longer time to reach the dynamic equilibrium of cation exchange reaction. Therefore, we set the reaction time at 1 hour in our experiment to ensure the complete cation exchange reaction.

Actually, although Figure 3c and figure S14 are the emission spectra of Tb-exchanged $NaGdF_4$ nanoparticles as a function of reaction time, they are obtained under different measurement conditions (such as different slit bandwidths).

Question 5: One of the emission bands of Eu doped materials was shown as 690nm in Fig S19, but 689 nm in Fig S17 and S32. Also, For Tb, the peak was shown at 489 and 541 nm in Fig S17, at 490 and 543 nm in figure S18b, and at 489 and 542 nm in figure S14. It is better to keep consistent to avoid the confusedness.

Response: We are grateful for this reviewer’s critical comment. As suggested, we have revised the figures (See Figure S14-19, S32) in supplementary materials.

Reviewer #2:

Han et al. reports on the synthesis and investigation of $NaGdF_4:Yb/Tm@NaGdF_4$ core shell NP’s doped with additional Tb ions through (most likely) cation exchange. This is very professional designed, brilliant written report. However from the scientific point of view, to my understanding, there is not much new. Nanoparticles showing anti-stokes emission upon upconversion are not new. As the authors themselves stated the core-shell strategy with or without energy migration has been exploited for different purposes and even tuning of the composition upon cation exchange has been shown at the nanoscale more than ten years ago (ref. 27-31). Thus the referee is wondering where are the new physical insights? That the upconversion emission between Ce^{3+} and Mn^{2+} in hexagonal-phased host material was obtained for the first time is perhaps true. To the referees opinion this does not justify publication in Nat. Commun.

Response: We would like to thank this reviewer for favorable comments on our presentation. As for the critical comments on the novelty of the work, we feel that there might be some misunderstanding, perhaps due to our unclear writing. Indeed, upconversion nanoparticles have been extensively studied in recent years. Various strategies have been explored to fine-tune upconversion emission. But these strategies mainly rely on the oil-based synthetic approaches (hydrothermal, co-precipitation, thermal decomposition, etc.). To obtain the multicolor upconversion nanoparticles, one has to synthesize a series of nanoparticles incorporated with different dopant composition, combination and concentration. The modulation of dopants often results in the variation of particle’s size, phase and morphology.

Indeed, the cation exchange has been reported for use in tuning the composition of lanthanide-doped nanoparticles. However, to the best of our knowledge, there are no examples reported on luminescence tuning of lanthanide-doped nanoparticles by the cation exchange approach. This is largely due to insufficient energy transfer from the sensitizers in the host lattice to the exchanged activators, mostly resting at the particle’s surface. Unlike the bandgap emission of quantum dots, upconversion luminescence from

lanthanide-doped nanocrystals is dominated by a cooperative effect of energy transfer, typically requiring a homogenous distribution of dopant ions in the host lattice.

Our work intends to report the following significant findings:

- The newly developed cation-exchange reaction provides a universal, just-in-time, post-synthesis strategy for preparing multicolour emitting upconversion nanocrystals.
- The fundamental constraints associated with size, phase and morphology variation of the nanocrystals by de novo synthetic techniques, can be mitigated by the process of cation exchange in water.
- Our approach exhibits a remarkably broad scope across a range of nanoparticle substrates.
- Using the cation exchange process, we observed, for the first time, the upconversion luminescence from Ce^{3+} and Mn^{2+} using hexagonal-phased nanocrystals.

We feel that these findings are significant as they lends new and exciting insights into chemical synthesis of multicolor emitting upconversion nanocrystals. The theoretical and experimental data presented herein suggest that the advantages associated with this cation exchange approach far outweigh its limitations. This work could open a gateway to access a myriad of upconversion nanomaterials that are relevant for fields as diverse as chemical sensing, biological imaging, photodynamic therapy, and anti-counterfeiting. This is the very reason why we have decided to submit this work to Nature Communications. And we hope this reviewer concurs after going through our clarification.

The obtainments made by the authors could simply be based on dissolution of the as-prepared NPs in (the still HCl containing solution) followed by growth of a Tb containing shell.

Can the authors exclude this? Did they try to find Cl in the NPs besides Tb? To the referees opinion this work cannot be accepted at this stage for publication in a high ranked journal.

Response: We thank this reviewer for the critical comment. With due respect, we feel that the reviewer misunderstood our experiments. HCl was used to remove the oleate ligand for yielding ligand-free nanoparticles. Subsequently, we precipitated the nanoparticles by centrifugation, following by washing nanoparticles with H_2O . Such procedure was carefully repeated for 5 times to ensure a complete removal of HCl in the solution. The pH value measurement in Figure RL1 shows that the colloidal solution is the same with the pure water, suggesting that no residual HCl is in the colloidal solution for the subsequent cation exchange reaction.

Figure RL1 (a) The photograph of the standard pH indicator shows the color chart corresponds to the different pH values. (b-d) The color change on the pH indicator strip after being dipped into HCl (1M), H_2O , and ligand-free NaGdF₄ nanoparticle solution after removal of HCl, respectively.

To validate that our synthesis is governed by a cation exchange process. We carried out ICP-MS analysis to examine Gd^{3+} content in the solution and Tb^{3+} content in the nanoparticles after treatment of $NaGdF_4:Yb/Tm@NaGdF_4$ nanoparticles with $TbCl_3$. As shown in Figure RL2, with increasing $TbCl_3$ concentration for cation exchange reaction, the Gd^{3+} content in the solution and Tb^{3+} content in the nanoparticles both significantly increased. The similar trend of ICP-MS test for Gd^{3+} content was observed in the $EuCl_3$ -treatment nanoparticle solution (Figure RL3). These data demonstrate that our synthesis is governed by the cation exchange process, rather than the deposition of the shell containing Tb^{3+} .

To further confirm our hypothesis, we carried out ICP-MS analysis to examine Gd^{3+} content in the solution after treatment of nanoparticles with Tb^{3+} under different reaction temperatures. We found that the Gd^{3+} content discharged from nanoparticles gradually increased with increasing the reaction temperature (Figure RL4), further substantiating the occurrence of cation exchange.

Figure RL2. (a) Inductively coupled plasma mass spectroscopic (ICP-MS) analysis of Gd^{3+} content released from the $NaGdF_4:Yb/Tm@NaGdF_4$ nanocrystals upon cation exchange with increasing Tb^{3+} concentration. (b) The ICP-MS data shows relative Tb^{3+} -to- Gd^{3+} content in the nanocrystals upon cation exchange with increasing Tb^{3+} concentration.

Figure RL3. (a) Inductively coupled plasma mass spectroscopic (ICP-MS) analysis of Gd^{3+} content released from the $NaGdF_4:Yb/Tm@NaGdF_4$ nanocrystals upon cation exchange with increasing Eu^{3+} concentration. (b) The ICP-MS data shows relative Eu^{3+} -to- Gd^{3+} content in the nanocrystals upon cation exchange with increasing Eu^{3+} concentration.

Figure RL4. Inductively coupled plasma mass spectroscopic (ICP-MS) analysis of Gd^{3+} content released from the $NaGdF_4:Yb/Tm@NaGdF_4$ nanocrystals after cation exchange with $TbCl_3$ as a function of reaction temperature.

It is important to note that the chlorides are highly hygroscopic and has a high solubility in water (*J. Chem. Educ.*, 2010, 87 (7), 727–729; *Chem. Soc. Rev.*, 2009, 38, 976–989). Therefore, the hygroscopic nature of chlorides prevent the generation of $TbCl_3$ shell. As suggested, we checked the elements of Tb^{3+} -treated $NaGdF_4:Yb/Tm@NaGdF_4$ nanoparticles using Energy-dispersive X-ray spectroscopy (EDS). As a result, no signal of Cl element could be detected from the nanoparticle (see Figure RL5 and Table RL1). Together, these results can further provide the evidence that the synthesis is governed by a cation exchange process.

Figure RL5. Energy-dispersive X-ray spectroscopy (EDS) of $NaGdF_4:Yb/Tm@NaGdF_4$ nanocrystals after cation exchange with $TbCl_3$.

Table RL1. EDS analysis report of NaGdF₄:Yb/Tm@NaGdF₄ nanocrystals after cation exchange with TbCl₃.

Element	Weight%	Atomic%
F	22.87	65.06
Na	4.57	10.74
Cl	0.00	0.00
Gd	41.39	14.23
Tb	4.98	1.69
Tm	12.37	3.96
Yb	13.83	4.32
Total	100.00	

On page two the authors said that "To further substantiate the occurrence of cation exchange, we carried out inductively coupled plasma mass spectroscopy (ICP-MS) analysis of the colloidal sample" Supplementary Fig. 11). So my question is. Did they investigated the solution or the "solid" sample? After ion exchange Gd ions should be dissolved in the solution.

Response: We tested Gd³⁺ ions both in solution and in the solid form. The results show that, with increasing amount of TbCl₃ (or EuCl₃), the amount of Gd ions increase in the solution as a result of ion exchange.

Reviewer #3:

A. With the burgeoning of actual and potential applications, upconversion nanoparticles (UCNPs) are among the most studied nano-objects presently. Several improvements have been made to their design in order to boost their quantum efficiency and their versatility. Yet, the synthesis of these nanoparticles remains involved and necessitate very strict control of experimental conditions rendering it time-consuming and, also, not easily reproducible. The merit of this work is to propose an elegant and simple way of designing series of UCNPs from an initial batch of gadolinium-based nanoparticles into which different lanthanide ions may be incorporated by cation-exchange under mild conditions.

A proof of concept is given for Tb(III) for which reaction time, temperature and ion concentration in the exchange solutions have been optimized. Next, the procedure is applied to three other trivalent lanthanide ions and to Mn(II), allowing the authors to demonstrate UC for the latter and Ce(III) in hexagonal NaGdF₄ nanocrystals for the first time.

B. Novelty. What is novel in the work is well described in the ms. The authors make use of their previous know-how on NaGdF₄ nanocrystals, in particular the fact that Gd(III) can act as energy reservoir and promotes energy migration, as well as described ion-exchange techniques in the synthesis of colloidal semiconductor nanocrystals. The merit of the authors is to have combined two concepts from two different fields to come up with the described results.

C. Altogether, the work has been conducted with great care. All necessary control experiments are presented, and experimental uncertainties are given. This is another high-quality work from this laboratory. I also note that the paper is not restricted to experimental results. First-principle calculations have been carried out to decipher the cation exchange mechanism, its energetics, and charge-transfer within the nanocrystals.

D. If the size, morphology and composition of the synthesized UCNPs are described adequately, nothing is indicated in the text or Supp. Mat. About the reproducibility of these parameters between different batches.

E. The conclusions are adequately supported by the data provided and should be very useful to scientists in the field.

F. As suggested improvements, I would like to see reproducibility data (see D above), as well as some comments/preliminary results on the applicability of the method to other types of UCNPs. Note: Figure S7: average sizes should not be given with so many significant digits, e.g. 21.34+-1.71 nm should read 21.3+-1.7 nm

G. References are appropriate

H. The ms is well organized and well written.

Response: We thank this reviewer's positive comments. To confirm the reproducibility of our synthesis strategy, we employed NaGdF₄:Yb/Tm@NaGdF₄ nanoparticles (26 mg) with an average size of 24.5 nm as the initial particle templates. After treatment with TbCl₃ and EuCl₃ (20 μmol), we found that the size and morphology of the resulting nanoparticles remain unchanged (Figure RL6).

In our manuscript, we have already demonstrated our cation exchange synthesis is applicable to the Tb-based nanoparticles (see Figure 4a). As suggested, we further studied cation exchange reaction in the other type of upconversion nanoparticles. Here, we carried out the cation exchange reaction by mixing NaYF₄:Yb/Tm@NaYF₄ nanoparticles with TbCl₃ (or EuCl₃). As anticipated, we did not observe an obvious change in the particle's size and morphology (Figure RL7). Together, these data suggest that our cation exchange is applicable to various types of upconversion nanoparticles.

As for the comment on "many significant digits", we have revised the figures (See figure S6-8) in supplementary materials according to the review's suggestion.

Figure RL6. Low-resolution TEM images and corresponding size distributions of the as-prepared NaGdF₄:Yb/Tm@NaGdF₄ (26 mg) nanoparticles, obtained before (a) and after (b,c) treatment with TbCl₃ (b, 20 μmol) and EuCl₃ (c, 20 μmol).

Figure RL7. Low-resolution TEM images and corresponding size distributions of the as-prepared NaYF₄:Yb/Tm@NaYF₄ (26 mg) nanoparticles, obtained before (a) and after (b,c) treatment with TbCl₃ (b, 20 μmol) and EuCl₃ (c, 20 μmol).

REVIEWERS' COMMENTS:

Reviewer #1 (Remarks to the Author):

the Authors have carefully addressed the list of questions raised by the referees, and I support its final acceptance by nature communications.

Reviewer #2 (Remarks to the Author):

The referee carefully checked the revised version of the MS as well as the responses letter. The responses given to my questions are of high quality and satisfying to me.

The referee would suggest the authors to add the text on page 4 plus Figure RL2, RL3 and RL4 to the supporting information of the MS. The referee is aware that anyway the content of the rebuttal letter will be inserted as additional supporting information but he would strongly recommend the authors to add this to the "normal supp. inf"

The referee recommends acceptance of the MS.

Reviewer #3 (Remarks to the Author):

My first report was positive and in my opinion the authors have thoroughly and adequately answered the questions raised by all the reviewers. The manuscript has been revised accordingly and I suggest acceptance.

Point-by-Point Response to the referees' comments (Manuscript # NCOMMS-16-13467A)

Reviewer #1:

The Authors have carefully addressed the list of questions raised by the referees, and I support its final acceptance by nature communications.

Response: We thank this reviewer's positive comments.

Reviewer #2:

The referee carefully checked the revised version of the MS as well as the responses letter. The responses given to my questions are of high quality and satisfying to me.

The referee would suggest the authors to add the text on page 4 plus Figure RL2, RL3 and RL4 to the supporting information of the MS. The referee is aware that anyway the content of the rebuttal letter will be inserted as additional supporting information but he would strongly recommend the authors to add this to the "normal supp. inf"

The referee recommends acceptance of the MS.

Response: We appreciate this reviewer's supportive comment. As suggested, we have added Figure RL2-RL4 and accompanying discussion into the Supplementary Information (See supplementary Fig. 34-36 and supplementary note 7).

Reviewer #3:

My first report was positive and in my opinion the authors have thoroughly and adequately answered the questions raised by all the reviewers. The manuscript has been revised accordingly and I suggest acceptance.

Response: We are grateful for this reviewer's favorable comments.